# The Pharmacorank Search Tool for the Retrieval of Prioritized Protein Drug Targets and Drug Repositioning Candidates According to Selected Diseases

**DOI:** 10.3390/biom12111559

**Published:** 2022-10-26

**Authors:** Sergey Gnilopyat, Paul J. DePietro, Thomas K. Parry, William A. McLaughlin

**Affiliations:** Department of Medical Education, Geisinger Commonwealth School of Medicine, 525 Pine Street, Scranton, PA 18509, USA

**Keywords:** protein database, search tool, prioritization algorithm, drug repositioning

## Abstract

We present the Pharmacorank search tool as an objective means to obtain prioritized protein drug targets and their associated medications according to user-selected diseases. This tool could be used to obtain prioritized protein targets for the creation of novel medications or to predict novel indications for medications that already exist. To prioritize the proteins associated with each disease, a gene similarity profiling method based on protein functions is implemented. The priority scores of the proteins are found to correlate well with the likelihoods that the associated medications are clinically relevant in the disease’s treatment. When the protein priority scores are plotted against the percentage of protein targets that are known to bind medications currently indicated to treat the disease, which we termed the pertinency score, a strong correlation was observed. The correlation coefficient was found to be 0.9978 when using a weighted second-order polynomial fit. As the highly predictive fit was made using a broad range of diseases, we were able to identify a general threshold for the pertinency score as a starting point for considering drug repositioning candidates. Several repositioning candidates are described for proteins that have high predicated pertinency scores, and these provide illustrative examples of the applications of the tool. We also describe focused reviews of repositioning candidates for Alzheimer’s disease. Via the tool’s URL, https://protein.som.geisinger.edu/Pharmacorank/, an open online interface is provided for interactive use; and there is a site for programmatic access.

## 1. Introduction

Proteins are currently being mapped to diseases in a comprehensive manner within open, online databases [1,2,3,4]. There is also an ongoing expansion of resources that document the associations of proteins with medications [2,5,6,7,8,9,10,11,12]. With the availability of these two types of information, proteins can serve as connection points between diseases and medications. Such connections have the potential to predict new indications for existing medications via the process of drug repurposing, also called drug repositioning [13].

To aid with the process of drug repurposing, estimates of the likelihoods that drug repurposing candidates could be effective new treatments have been made using various approaches. These approaches include network-based approaches, text-based approaches, and semantics-based approaches, as reviewed by Xue et al. [14]. A central component of some of these approaches is a gene prioritization algorithm that uses similarity profiling, and such algorithms offer demonstrated applicability to the prioritization of proteins associated with diseases [15]. The proteins’ priority or rank scores, in turn, mirror the likelihoods that they may be useful as medication targets for the treatment of a selected disease [16]. Since the priority scores of the protein targets can be linked to their associated medications, the result can be the uncovering of novel predictions regarding which medications may be most effectively repurposed [17].

Methods that prioritize proteins involved in diseases may use the presence of co-occurring words or database terms as part of the prioritization algorithm. Consider, for example, the use of co-occurring database terms in the PolySearch tool [18]. We previously implemented a tool called KB-Rank that considers the co-occurrences of a diverse set of functional annotations to prioritize proteins associated with selected diseases [19]. More example methods that prioritize protein–disease datasets using protein functions include ToppGene [20], TargetMine [21], and network methods [22,23]. Examples of the integration of gene prioritization methods into an online search tool with the goal of aiding drug repositioning efforts have been implemented in RepurposeDB [24] and Project Rephetio [25].

To further enable drug repurposing efforts, we present the Pharmacorank search tool. To prioritize the candidates, an unsupervised gene prioritization algorithm is implemented, which may be classified as a similarity profiling or a data fusion method [26]. This method utilizes the diverse set of protein functions and annotations that are available via the UniProt database [3].

To validate and benchmark the search tool’s accuracy, we calculated the percentages of medication–protein target pairs that were already known to be involved in the disease’s treatment for each priority range, and we termed these values the pertinency scores. An optimal correlation was sought between the priority scores and the pertinency scores. The equation of best fit then served as a predictive model that takes the priority score of each of the protein targets and predicts the pertinency score. We interpret the pertinency scores as being estimates of the likelihoods that the medications associated with the protein targets will be clinically relevant in a selected disease’s treatment.

With the goal of helping to spur drug repositioning efforts within the greater scientific community, the predictive model between the priority score and pertinency score was applied to all diseases described in Disease Ontology (DO) [27], and drug repositioning candidates were made available for each of the diseases described in DO that map to an ICD-9-CM or ICD-10-CM code. To highlight the results, a few of the drug repositioning candidates with the highest predicted pertinency scores across all the diseases are further described. We also provide focused manual reviews of the drug repositioning candidates that are identified as possible future treatments for Alzheimer’s disease (AD).

## 2. Materials and Methods

### 2.1. Overview

The overall steps of the method to obtain prioritized proteins and medications to inform drug repurposing studies may be summarized as: (1) retrieve the proteins associated with each identified disease; (2) rank the proteins with priority scores calculated using protein functions; (3) obtain the correspondences between medications and proteins along with correspondences between medications and diseases from public databases to determine whether each protein interacts with a medication already known to treat the disease; (4) perform validation studies regarding the contribution that each type of protein function has in generating a priority score that discerns whether a protein interacts with a medication that is already used to treat the disease; (5) derive a predictive, quantitative relationship between the priority score and pertinency score, where the pertinency score is the percentage of protein targets known to interact with medications that are already known to treat the disease; (6) apply the resulting predictive relationship between the priority score and pertinency score to all proteins and medications associated with the disease; (7) identify a recommended pertinency score threshold for the end-user; and (8) review the protein targets and medications that have high predicted pertinency scores for consideration in future drug repositioning studies.

### 2.2. Assemble Protein–Disease Datasets (Overall Step 1)

A set of proteins associated with each disease was obtained based on information from multiple sources. Known correspondences were retrieved based on the Online Mendelian Inheritance in Man (OMIM) phenotypic descriptions [1] for a total of 4646 correspondences. Further, an additional 1798 correspondences between diseases and proteins were obtained using the Kyoto Encyclopedia of Genes and Genomes (KEGG) disease name assignments [2]. The integrated protein–disease datasets from DisGeNET [28] added an additional 54,226 correspondences. The total number of protein–disease correspondences was therefore 60,670. With nonredundant protein–disease datasets derived collectively from these multiple sources, the result was a comprehensive dataset for each DO entry.

#### 2.2.1. Implement the Prioritization Algorithm Using Protein Functions (Overall Step 2, First Part)

The functions used in the prioritization algorithm were retrieved from UniProt files and from coordinating resources that describe protein function. The types of functions include UniProt keywords, Gene Ontology terms, Enzyme Commission (EC) numbers, InterPro assignments, SUPERFAMILY assignments, small molecule interaction assignments from Chemical Entities of Biological Interest (ChEBI) [11], and UniProt residue features.

There were three types of annotations used from Gene Ontology (GO): molecular function, cellular component, and biological process. GO terms were obtained through UniProt GOA, which had granular GO annotations and excluded those higher up within the GO hierarchy when identified by the same technique [29,30]. All ChEBI entries except those that mapped to ChEMBL entries were utilized as function assignments. Broadly, the functions used here are functional characteristics that are shared among two or more proteins. Functional characteristics that could only possibly be attributed to one protein were excluded. For example, point mutations were excluded as functions since a point mutation would not be shared with other proteins.

#### 2.2.2. Calculate the Priority Scores (Overall Step 2, Second Part)

After the identification of the proteins of each protein–disease dataset, the priority scores of the proteins and their associated medications were calculated. An outline of the prioritization algorithm is shown in Figure 1. Raw priority scores were first calculated separately according to each type of function. For this purpose, the total number of proteins in the protein–disease dataset with a specific function was found, and this number was assigned to the specific function. A protein’s raw priority score according to the type of function was then the sum of the numbers assigned for each specific function for which the protein was known to have. Mathematically, the raw priority score of a protein using a type of function is equivalent to the dot product of two equal length 1D arrays, rho and mu. The array mu, μ, has an entry for each of the specific functions found among the proteins in the protein–disease dataset with the value that is equal to the number of UniProt accession codes (proteins) in the protein–disease dataset that had that specific function. Each protein in the protein–disease dataset also had its own associated binary array rho, ρ, which consisted of one if the specific function was attributed to that protein or zero otherwise. The formula for the raw priority score for a protein regarding a type of function was calculated as the dot product of ρ and μ:$${Pscore}_{raw}^{FuncType}=\rho \cdot \mu$$

To calculate the raw priority score, a Python dictionary was created, where the keys were the unique identifiers of the specific functions in the protein–disease dataset and the values were the corresponding total numbers of proteins that had each specific function. The raw priority score for a protein regarding a type of function was then the sum of the values associated with each of the specific functions that were attributed to that protein. This is numerically equivalent to the dot product but avoids what we found to be a more computationally intense and more error-prone task of creating and multiplying the 1D arrays ρ and μ

The raw priority score of each protein was normalized using the total number of proteins in the protein–disease dataset and the total number of distinct specific functions of the type of function under consideration that were represented in the protein–disease dataset. The normalization factor eta, η, was obtained with the following formula:$$\eta^{FuncType}=\sqrt{p^{2}+a^{2}}$$ where the variables *p* and *a*, respectively, are the number of proteins retrieved in the protein–disease dataset and the number of distinct specific functions represented in the protein–disease dataset with reference to the type of function.

We inferred that *η* was proportional to the combined error associated with the measurements of the number of proteins in the protein–disease dataset and the numbers of unique specific functions represented for proteins in the protein–disease dataset. Based on this inference, the normalization then follows a standard procedure of dividing by the total error [31], where the total error was obtained by adding the contributing errors in quadrature. Since the numbers of proteins and functions can vary greatly across diseases, the normalization factor η helped to ensure that the values of the priority scores were on a comparable scale for the different diseases.

The rate at which functional features were assigned to proteins varied greatly according to the type of function. We therefore introduced the second normalization factor beta, *β*, which considered the average number of functions per protein for each type of function separately. Upon application of both normalization factors, the resulting priority score of a protein was calculated with the following formula:$${Pscore}_{norm}^{FuncType}\frac{{Pscore}_{raw}^{FuncType}}{\eta^{FuncType}\cdot \beta^{FuncType}}$$

The final normalized priority score of a protein for a given disease was then the average of the normalized priority scores calculated when using each type of function separately. In the formula, n is in the number of different types of functions considered, which was 9. As described above, the types of functions were UniProt keywords, the three types of Gene Ontology terms (molecular function, cellular component, and biological process), Enzyme Commission (EC) numbers, InterPro assignments, SUPERFAMILY assignments, small molecule interactions from ChEBI, and UniProt residue features. $${Pscore}_{norm}^{Average}\frac{\sum {Pscore}_{raw}^{FuncType}}{n}$$

### 2.3. Select Diseases Used for the Validation Studies (Overall Step 3)

To select the diseases from Disease Ontology that were to be used in the validation studies, the following procedure was implemented. The correspondences between medications and indications were retrieved from three sources: Medication-Indication (MEDI-C) resource [32], DrugCentral [12], and ChEMBL [7]. All medications were classified as being on the market with a status of being in phase 4, and each medication did not have a flag indicating it had been withdrawn from the market as per the annotations in DrugCentral. For each indication, the diagnostic code(s) from the International Classification of Diseases, Clinically Modified, was obtained from the 9th and 10th editions, which are referred to, respectively, as the ICD-9-CM and ICD-10-CM codes. The DO term description that corresponded to each diagnostic code was subsequently obtained using the correspondences that are available in the DO OBO file.

Each of the diseases analyzed were required to have at least one medication currently on the market for the disease’s treatment, which interacted with a protein in the corresponding protein–disease dataset. To ascertain whether a retrieved protein interacted with a known medication for a selected disease, the list of medications for that disease was compared to the list of medications associated with the protein obtained from ChEMBL [7]. Diseases that were nonspecific such as “cancer” or “skin disease” were excluded; these are listed as the vague diseases in the Appendix A.

### 2.4. Evaluate the Contributions of the Types of Functions to the Priority Score Accuracy (Overall Step 4)

To evaluate the utility of the priority score for discerning medication–protein target pairs that are clinically useful for the treatment of a selected disease, receiver operator characteristic (ROC) curves were generated. The positives were defined as those proteins in the protein–disease dataset that interacted with one or more medications currently known to treat the disease. The rest of the proteins in the protein–disease dataset were the negatives.

Fixed values of the priority scores that corresponded to each protein were used as thresholds. The sensitivity and specificity values at each threshold for each of the different protein–disease datasets were obtained. The area under the curve (AUC) for each ROC for each protein–disease dataset was calculated, and an average AUC across the protein–disease datasets was obtained. The AUC calculations were performed using a Python script developed for this purpose. For the analyses, each protein–disease dataset was required to have 30 or more proteins to ensure there was ample data to estimate the sensitivity and specificity measurements at each threshold. Each protein–disease dataset used for validation was also required to have at least one protein that was known to be the target of a medication used to treat the corresponding disease.

To identify medications that may be repositioned to treat a selected disease, we first identified those medications already known to treat the disease. A match was sought between the ChEMBL medication identifier and the medication identifier in MEDI-C or DrugCentral. Matching was carried out by one of the following ways: match the identification codes between RxNorm Ingredient ID [6] of MEDI-C to ChEMBL ID with the normalized names for clinical drugs (RxNorm) database, table ‘RXNCONS’O; or match the generic medication name with the text match. Since the MEDI-C resource contained medication–indication pairs for which the medication was a combination of two or more drugs, each drug of each combination was connected separately.

### 2.5. Relationship between the Priority Score and Pertinency Score (Overall Step 5 and 6)

The relationship between the priority score and percentage of proteins that interact with medications currently known to treat a selected disease, referred to as the pertinency score, was assessed. To generate a plot of priority scores versus pertinency scores, six equal intervals were considered along the full range of priority scores. Six cross-fold validation sets, each containing 16.6% of the diseases of the full validation set, were generated for each of the intervals. For each fold, the percentage of proteins targeted by medications known to treat the selected disease was calculated. A scatter plot of the average of these percentages, called the pertinency scores, versus the average priority scores was generated. A fit of the scatter plot was constructed using a weighted, second-degree polynomial using the lm package R [33]. The average inverse of the variances of the pertinency scores was used as a weighting factor in the scatter plot [34]. These variances were based on the six values that were obtained using the six folds of the validation set.

### 2.6. Identification of a Threshold for the Pertinency Score (Overall Step 7)

To provide the end-user with a threshold for the pertinency score, tests were conducted using the drug to disease correspondences from MEDI-1, which was created in 2013, and the drug to disease correspondences from MEDI-2. The goal was to empirically identify the pertinency score range(s) of the drug to protein to disease tuples that were identified using MEDI-2 data but were absent when using MEDI-1 data. The study protocol was run using only the MEDI-1 data and only the MEDI-2 data separately.

We then examined the number of new entries for different pertinency score ranges to observe empirically where the new entries fell. To normalize the numbers for each range, the number of new drug/protein/disease tuples for each pertinency score range were divided by the total number of protein targets that fell within each corresponding pertinency score range. We then plotted a bar chart of these ratios to empirically observe the cut-off point, where there was a large increase in the estimated likelihood of success.

### 2.7. The Retrieval of Results on the Pharmacorank Site (Overall Step 8)

After completing the validation and prediction studies, we expanded the diseases considered and further applied the resultant prediction mode based on the second-degree polynomial fit of the priority score versus pertinency score. For the presentation of the predicted pertinency scores on the Pharmacorank website, analyses of all diseases described in Disease Ontology that mapped to either an ICD-9-CM or ICD-10-CM code were performed. The pertinency score was calculated for each protein in each protein–disease dataset regarding all corresponding proteins within Swiss-Prot, which does not include TrEMBL.

## 3. Results

### 3.1. Validation Studies of the Priority Score

We first tested the accuracy of the priority score regarding its ability to discern, among all the proteins of a protein–disease dataset, those that were targets of medications already known to treat the disease. ROC curves were obtained for each disease, where the positives are the known medication targets and the negatives are the rest of the proteins within the protein–disease dataset. All protein–disease datasets with thirty or more proteins that had one or more proteins that were identified as interacting with a currently used medication for the disease were included. There were 513 diseases that met these criteria. The list of these 513 diseases in the validation set is provided in the Appendix A. The results presented here are based on the 3 August 2022 timestamp of the UniProt/Swiss-Prot data.

The effect on the AUC values of using each type of function separately for the calculation of the priority scores is shown in Table 1. We observe that when using the SUPERFAMILY or ChEBI assignments only, their resultant AUC values were not significantly higher than an AUC of 0.5, which is the value that corresponds to no discrimination. These two types of functions were therefore removed from the formulation of the priority score. The average AUC value obtained when all types of functions except SUPERFAMILY and ChEBI were retained was 0.68661.

### 3.2. Predictive Relationship between the Priority Score and Pertinency Score

We modeled the relationship between the priority score of a protein in a protein–disease dataset and its pertinency score. As described in the methods, six equal intervals along the priority score range were identified, and the average priority score for each of the intervals was calculated. Then, for each priority score interval, the fraction of proteins in the interval that were targets of medications currently used to treat the disease, referred to as the pertinency score, was calculated. To obtain the average and standard error estimates for the priority scores and pertinency scores, a six-fold cross-validation was implemented as described in the methods.

A weighted least-squares fit with a second-degree polynomial was found to have the following equation: y = 1.399x^2^ − 0.110x + 0.015. The correlation coefficient for the fit was 0.9978, which indicates a highly predictive relationship, as shown in Figure 2. The polynomial equation for the fit was subsequently applied to estimate the pertinency score of a protein for a selected disease given its priority score for that disease.

The pertinency score is interpreted as an estimate of the probability that the medication would be relevant in the treatment of the corresponding disease. To be clinically useful in the disease’s treatment, the medication would need to oppose the aberrant function of the protein, which contributes to the disease. This determination is left to the user. The tool estimates the strength of the association, but the user must then determine if the drug can theoretically be indicated or contraindicated. For example, if a protein’s function increases in the disease state and the medication inhibits the corresponding protein function, the medication would likely be clinically useful in the disease’s treatment. If the medication further increases the aberrant protein function or the medication exacerbates a loss of function that is associated with the disease, the medication would be relevant as a possible contraindication. The direction of the effect of the medication and the direction of the aberrant protein function in the disease mechanism therefore need to be ascertained to know whether the drug’s effect would be useful in the control the disease [35].

### 3.3. Estimation of an Empirical Threshold for the Pertinency Score

As described in the methods, the analyses were conducted separately using only known drug to disease correspondences from MEDI-1 and then using only the drug to disease correspondence from MEDI-2. The pertinency scores of the drug/protein/disease tuples found from the MEDI-2 analyses but not from the MEDI-1 analyses were used to estimate the pertinency score range(s) where the tuples with a high likelihood of successful development would lie in the future.

Figure 3 shows a bar chart with the ratios of the pertinency scores of new drug/protein/disease tuples to the number of protein targets for each of the corresponding pertinency score range bins. For the range of 0–0.1, there were 4790 new tuples and 100,369 protein targets, which gave a ratio of 0.048. For the range of 0.1–0.2, there were 1231 new tuples and 7019 protein targets, giving a ratio of 0.175. For 0.2–0.3, there were 96 tuples and 625 protein targets, giving a ratio of 0.154. For 0.3–0.4, our analysis detected no new tuples. Within 0.4–0.5, there were 2 tuples and 75 protein targets, giving a ratio of 0.027. Based on the large jump when moving to the 0.1–0.2 bin, and a review of the rest of the chart, we infer that a threshold of 0.1 or above for the pertinency score captures new drugs effectively while eliminating most of the protein targets from consideration. We make the inference that new drug repositioning candidates would also likely have pertinency scores above the 0.1 threshold and have the highest likelihoods of ultimately becoming useful drugs.

### 3.4. Three Illustrative Examples of Repositioning Candidates

We reviewed the examples of the top repositioning candidates with the highest priority scores and therefore the highest predicted pertinency scores across all the diseases analyzed. The total number of unique diseases was 4041. The idea is that by considering those candidates with the highest predicted pertinency scores across all the diseases, a focus would be placed on those most likely to be clinically relevant. We selected three medications among the top predicted repositioning candidates to gain insight as to whether these candidates would likely be clinically useful. Additionally, these examples help to illustrate what one would expect to encounter when carrying out such reviews.

The main goal of each individual candidate review was to identify cases where the drug has an opposing effect to that of the aberrant function of the protein. This boils down to finding cases where there is a gain of function of the disease protein and the drug inhibits the corresponding function. Alternatively, it corresponds to cases where the protein has a loss of function variation, and the drug increases the corresponding function of the protein. Cases where the drug enhances a gain of function of a disease or where the drug inhibits the function of a protein that had a corresponding loss of function would likely be contraindicated.

Three proteins and repositioning candidates selected from across all the diseases are listed in Table 2. A top repositioning candidate is sotorasib for the possible treatment of linear sebaceous syndrome. Sotorasib inhibits GTPase KRas. Consider that while linear sebaceous syndrome is usually associated with a benign skin lesion, more severe phenotypes such as malignant tumors may also manifest [36]. Linear sebaceous syndrome can also be associated with cerebral, ocular, or skeletal defects, which together are referred to as Schimmelpenning syndrome. The mutant GTPase KRas of the disease state has a higher proportion of HRAS-GTP activity than that found in wild-type cells [36]. The inhibitor sotorasib may therefore offer a treatment to reduce the relatively high activity seen of GTPase KRas in the disease state [37].

For the review of the second candidate, consider GM2 gangliosidosis, which is progressive lysosomal storage disease marked by the accumulation of GM2 gangliosides in neuronal cells. This condition is caused by loss of function variations in beta-hexosaminidase subunit beta protein, and the phenotype of GM2 gangliosidosis is indistinguishable from that of Tay–Sachs disease. A possible repositioning candidate is pyrimethamine, which is a pharmacological chaperone (PC) that can stabilize the conformation of the mutant protein [38]. This allows the protein to pass quality control, avoid degradation, and continue to function. Although pyrimethamine can cross the blood–brain barrier and increase the beta-hexosaminidase activity, clinical trials have described limited impact on the manifestations of the disease in the central nervous system [39,40].

A third repositioning candidate is tolcapone for the possible treatment of dengue hemorrhagic fever. Tolcapone inhibits the serine protease function of NS3. The dengue virus type 2 NS3 protein is one of the cleavage products of a large 3391-amino-acid glycoprotein from the dengue virus. The combination of several proteins into one large genome glycoprotein that is subsequently cleaved into functional smaller proteins has apparently enabled all the proteins of this genome glycoprotein to have high priority scores since all the functional annotations of all the proteins (cleavage products) that make up the glycoprotein would have been used in the prioritization algorithm. Further, given that there are multiple strains of the dengue virus, which means repeats of similar glycoproteins of the dengue virus are described in UniProt, these glycoproteins have high calculated priority scores. Nonetheless, the NS3 protein is described as being a target of tolcapone, and tolcapone is reported as a hit from a high-throughput screening with a Ki value range of 0.61–1.25 μM [41,42]. These results point to possible treatment of the viral infection through the possible derivation of tolcapone as a hit compound and the subsequent steps required for drug development.

### 3.5. Examples of Repositioning Candidates for Alzheimer’s Disease

A list of three protein targets and their associated medications to consider for repositioning studies for AD is provided in Table 3. One repurposing candidate is insulin, which binds to the insulin receptor. Insulin is used to treat type 1 diabetes mellitus and type 2 diabetes mellitus [43]. The insulin receptor has a high predicted pertinency score, and insulin has demonstrated disease-modifying activity that opposes the disease mechanism. A recent study by Keller et al. found that intranasal insulin has demonstrated clinical benefit based on a phase 2 clinical trial [44]. Relative to the control group, the insulin-treated group showed beneficial changes in CSF immune/inflammatory/vascular markers. Beneficial changes in cognition, brain volume, and both amyloid and tau concentrations were also observed. The authors conclude that intranasal insulin may promote a compensatory immune response that is associated with the therapeutic benefit.

A second repurposing candidate is riluzole, which has the ability to reduce the alpha-synuclein protein aggregation seeds [45]. The current indication for riluzole is amyotrophic lateral sclerosis [43]. There is evidence that the pathology of AD is linked to alpha-synuclein via multiple mechanisms that include asymptomatic accumulation of Aβ plaques and tau hyperphosphorylation [46]. A clinical trial demonstrated a strong correlation between riluzole treatment, cognitive measures, and brain metabolism in those with AD. The changes in brain metabolism included a slower rate of cerebral glucose metabolism decline [47,48]. Further, in a mouse model, riluzole impacted some immune-related pathways that are implicated in AD [49].

We found that angiotensin-converting enzyme 1 (ACE) has a relatively high pertinency score. Some evidence suggests that ACE2 inhibitors are associated with a slower rate of cognitive decline [50], but this evidence appears to be inconclusive. Alternatively, evidence also points to a protective effect of angiotensin-converting enzyme 1 against AD. Specifically, the function of ACE within the cerebrum is needed for a protective effect in AD, and the associated function of ACE is possibly independent of its contributions to the control of blood pressure [51]. In addition to the inhibitors retrieved here via the ChEMBL mappings to ACE, we observe that there are known activators of angiotensin-converting enzyme 2 (ACE2) [52,53]. Additionally, we observe that the neuroinflammation pathways involved in neurodegeneration can have associated decreases in ACE2 activity and increases in ACE1 activity [54]. Further, in a mouse model of AD, the ACE2 activator diminazene aceturate (DIZE) reduced the levels of Aβ1-42, hyperphosphorylated tau, and pro-inflammatory cytokines in the brain [55]. DIZE is a veterinary drug used to treat blood-transmitted protozoan parasites such as trypanosoma, and it has also been used to treat human trypanosomiasis without major toxicity [56]. This example highlights the need to review each target that has a high pertinency score and see which medications, regardless of whether they are currently represented in the ChEMBL mappings, may oppose the overall disease pathway.

## 4. Discussion

### 4.1. Pharmacorank’s Possible Role in Enabling Drug Repositioning

Drug repositioning can improve treatment outcomes and reduce the cost of drug development [57]. Marketed drugs have already been through clinical trials, so the number of trials that would be required during the drug repositioning process would be reduced. Such a reduction can save approximately 2 years of time and 40% of the overall cost of drug development [58].

Computational approaches for drug repositioning prioritize their identified candidates based on their estimated likelihoods of success [17,59]. The estimates of the success rates can be made using the known, clinically used medication–indication pairs. These estimates can further aid in the selection of the drug repositioning candidates that are to be moved forward through drug development. This information has demonstrated importance for modeling purposes [60] and validation purposes [61].

As described through our manual reviews, the results of searches across different diseases with Pharmacorank can be collated, and drug repositioning candidates with the highest predicted pertinency scores can be identified for review. These candidates may constitute the lowest hanging fruits, where a focus on further drug discovery and development may be placed [62]. Further, when there is identification of protein targets with a determined three-dimensional structure or accurate structural models, computational approaches that use structural information, such as for rational drug design [63], docking, and/or virtual screening, may be readily applied [64].

We anticipate that the Pharmacorank search tool will complement other open technologies that are available to aid in the identification of new possible treatments for orphan diseases. There is a need for software applications that identify drug repositioning candidates for orphan diseases [65,66]. Our findings regarding the repositioning candidates for the possible treatments of linear nevus sebaceous syndrome, dengue hemorrhagic fever, and GM2 gangliosides highlights examples for orphan diseases. The need to fully bring to light easily searchable and viable drug repositioning candidates for orphan diseases is apparent. The long-term goal would be to improve clinical outcomes for these conditions. The identification of new uses for old drugs may also add value by enabling a drug to enter into a new market for the treatment of an orphan disease, which may extend the patent life of the drug [67,68].

### 4.2. Relation to Other Tools for Drug Repositioning

Drug repositioning methods span a variety of different experimental and computational approaches [67,69,70], which may be grouped according to whether they predict new interactions between medications and proteins or just prioritize the proteins involved in the disease. Those that predict new physical interactions between medications and protein targets may validate their results by testing the predicted interactions using ligand-binding assays [71]. In contrast, methods that prioritize known medication–protein target pairs may validate predictions using cross-validation studies or by evaluating their effects on disease phenotypes using animal models [72]. In the absence of performing clinical trials, both types of methods may be validated using evidence from the literature that describes the possible clinical usefulness of medications for a selected disease.

The application of Pharmacorank for drug repositioning may be classified into the latter group of methods, since the priority scores of medication–protein target pairs are described, and no new physical interactions between protein targets and medications are predicted. Comparable methods include ToppGene [20], TargetMine [21], a network method by Emig et al. [22], and DrugNet [73]. ToppGene discusses drug repositioning candidates in the context of evidence from literature references that support their plausibility. DrugNet takes validation a step further by performing cross-validation of the predicted medication–indication pairs using data from clinical trials. The use of DrugNet reports an AUC value of 0.836 when the positives were medication–indication pairs found to be in clinical trials and the negatives were randomly selected drugs.

Regarding the network method reported by Emig et al., the positives were those proteins known to be targeted by a medication in a clinical trial for the treatment of the disease and the negatives were randomly selected protein targets [22]. The Emig study reports AUC values that range between 0.63 and 0.93 for different diseases. In a study by Kissa et al. [74], algorithmic approaches for unsupervised prioritization of drug repositioning candidates were also described. The validation sets include positives that were approved medication–protein target pairs for each disease in question [74]. The negatives were random pairs of drugs with targets. They report an overall AUC value of 0.84 for the discovery of medication–indication pairs using the Pointwise Mutual Information algorithm.

For the Pharmacorank search tool, the following validation approach was undertaken: the priority score was used to discern proteins targeted by medications used to treat the disease from all the other proteins associated with the disease. The Pharmacorank search tool similarly falls into the category of methods that utilize AUC values for validation, and it therefore falls into the category of methods that use both sensitivity- and specificity-based validation (SSV) as described by Brown and Patel [75]. Here, the priority score is used to discern proteins targeted by medications used to treat the disease from all the other proteins associated with the disease. This was carried out based on the coordinated mapping of the medication–indication pairs in MEDI-C and DrugCentral with the medication–protein target pairs in ChEMBL. For the described previous studies, we observe that the values for AUCs are higher than those described in the current study. One difference in the approaches is that random proteins were part of the comparison sets of the previous described studies and not all these random proteins were deemed to be directly involved in the disease.

For example, in the Emig study, all the proteins of the comparison set have a score that relates each protein to the disease but not all the proteins were deemed to be directly involved in the disease. In our studies, we note that our negatives do not include many random proteins not directly associated with the condition. Specifically, our negatives are proteins directly involved in the disease but are not known to interact with a known drug to treat the disease. Since we are contrasting the functions of proteins directly involved in the disease that interact with medications known to treat the disease versus the functions of protein directly involved in the disease that do not interact with medications known to treat the disease, we infer that the method enabled a more precise identification of those functions most relevant to make the protein a viable target whose corresponding function could be modified as part of the disease’s treatment.

We interpret the AUC values obtained here as being more relevant and applicable to drug repositioning. This is because discerning the most clinically useful targets among the proteins known to be directly involved in the disease is likely what will be most useful in practice. Drug discovery and development projects focus on the proteins known to be involved in the disease and then try to figure out which of those will be the most effective drug targets.

### 4.3. Significance of the Relationship between the Priority Score and Pertinency Score

One of the take home messages from the analyses is that there is a predictive phenomenological relationship between the priority score and pertinency score. Knowledge of the set of proteins involved in a disease along with their normal biological functions can be used to quantitatively predict the likelihood that a given protein and its associated medications would be relevant in the treatment of the selected disease.

We note that in the relationship between the priority score and pertinency score, as the priority score reaches relatively high values, the standard deviations of the pertinency scores increase. One reason for this observation was noted earlier, which is that some proteins are part of polyproteins, such as what is typically found in viruses. These proteins will receive high priority scores since all the protein functions would be attributed to a single UniProt entry that represents the entire polyprotein prior to cleavage. Furthermore, since there are multiple viral strains represented in UniProt, protein functions would be repeated multiple times in the prioritization algorithm, thus artificially increasing the corresponding priority scores. Of note is that the predictive relationship between the average priority score versus the average pertinency score holds.

The observation that this strong relationship between the priority score and pertinency score exists is significant since the data assembled in UniProt and the data from the clinical trials of medications are of disparate origins. The information in UniProt regarding the characterization of the normal functions of proteins is independent of the testing and validation of medications brought to market via the required clinical trials.

When considering the selection of drug repositioning candidates to consider for a selected disease, we recommend starting with those with the highest predicted pertinency scores that are likely to oppose the disease pathway. In many cases, the pertinency score may not reach above 10–20%, but those with the highest predicted pertinency scores are recommended to be the ones to reviewed first. There are many factors that contribute to this low rate such as, simply, not all the drugs have been successfully developed yet so the percentage of proteins that are targeted by clinically useful medications has not achieved its maximum value.

As described in the results section, we recommend that the end-user consider a threshold of 0.1 for the pertinency score. We further note that the current success rates for drug development from the downstream points of the beginning of the clinical trial to the point of receiving marketing approval has remained about 10–20% [76]. Although the pertinency score is not on the same scale, we surmise that a drug repositioning candidate with a pertinency score above 10% should garner attention as these are predicted to have likely paths to success that is possibly better than many of the drugs that have reached the stage of entering clinical trials.

Now, the Pharmacorank site reveals that there are hundreds of drug repositioning candidates across a wide range of conditions that meet the 0.1 threshold. These candidates may be particularly ripe for the clinical trials that are required in their development for the selected conditions. As an additional repurposing example candidate that meets the 0.1 threshold, we observe that probucol has a pertinency score of 0.102 through its interaction with vascular cell adhesion protein 1 in the condition multiple sclerosis. We note that probucol has been shown to reduce neural cell apoptosis after cellular injury [77]. This example further highlights the potential of identifying viable drug repositioning candidates for future drug development studies as guided by the threshold value of the pertinency score.

### 4.4. Access

The web interface for the Pharmacorank search engine is available at the URL http://protein.som.geisinger.edu/Pharmacorank/. Automated monthly updates of the searchable content and the predictive model are enabled based on the routine updates of the information in UniProt. All analyses are pre-run prior to making them available on the website to make the queries fast since it is then just a matter of looking up the pre-computed results. For interactive queries, an autocomplete tool identifies the corresponding Disease Ontology term as a disease name is being typed. For each protein and drug retrieved, links are provided to the corresponding entries in UniProt and ChEMBL for further information. Programmatic access is enabled to retrieve the results of precomputed searches via the URL http://protein.som.geisinger.edu/Pharmacorank/Downloads/.

## 5. Conclusions

The Pharmacorank search tool provides a means to retrieve protein medication targets and their associated medications that are either known or predicted to be relevant in the treatment of a selected disease. The results of searches are prioritized using an objective algorithm that considers each protein target’s complement of functions. The functions are derived from the broad collection of function descriptions in UniProt. Different types of functions are collectively used in the formulation of the priority scores. We find a quantitative, predictive relationship between the resulting priority score of a medication–protein target pair and its probability of being clinically relevant in the treatment of the selected disease.

To facilitate drug repositioning efforts across a wide range of diseases, the disease terms and phrases described in Disease Ontology were analyzed. The medications associated with the retrieved proteins were considered as drug repositioning candidates if they were not yet used to treat the queried disease and were likely to oppose the disease mechanism. We anticipate that the drug repositioning candidates described here and those found subsequently through the updated search tool will ultimately be clinically relevant for their predicted indications, which may lead to cost savings and a reduction in disease burden. An emergent feature of the search tool is that the repositioning candidates most likely to be clinically relevant across a wide range of diseases could be readily identified since the priority scores and pertinency scores are normalized across all the diseases analyzed.

## Figures and Tables

**Figure 1 biomolecules-12-01559-f001:**
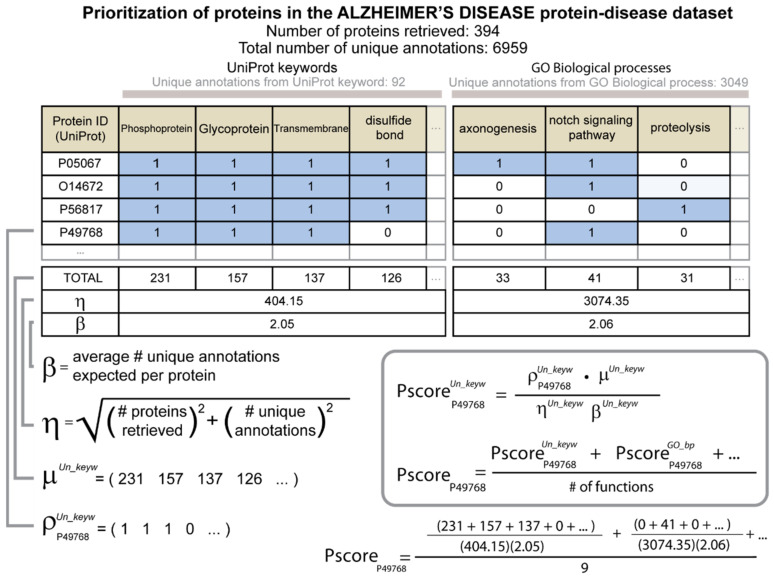
Diagram describing how priority scores are calculated. The protein–disease dataset for AD is used in the example. The proteins are identified by their corresponding accession codes in UniProt. A unit matrix is then created where the rows are the UniProt entries, and the columns are specific functions. The 1D array rho, ρ, is a binary array that represents the presence or absence of each function for a given UniProt entry. The 1D array mu, μ, holds the total number of UniProt entries for each function among the proteins of the protein–disease dataset. The dot product of rho and mu produces a raw priority score. The factor eta, η, is one of the variables that is used to normalize the raw priority score. Eta is calculated by summing in quadrature the total number of UniProt entries and the total number of functions associated with the disease for a given type of function. A second normalization factor beta, β, is the average number of unique functions per protein of a given type of function. The priority score of a protein is the mean of the normalized priority scores that were calculated separately using each of the different types of functions.

**Figure 2 biomolecules-12-01559-f002:**
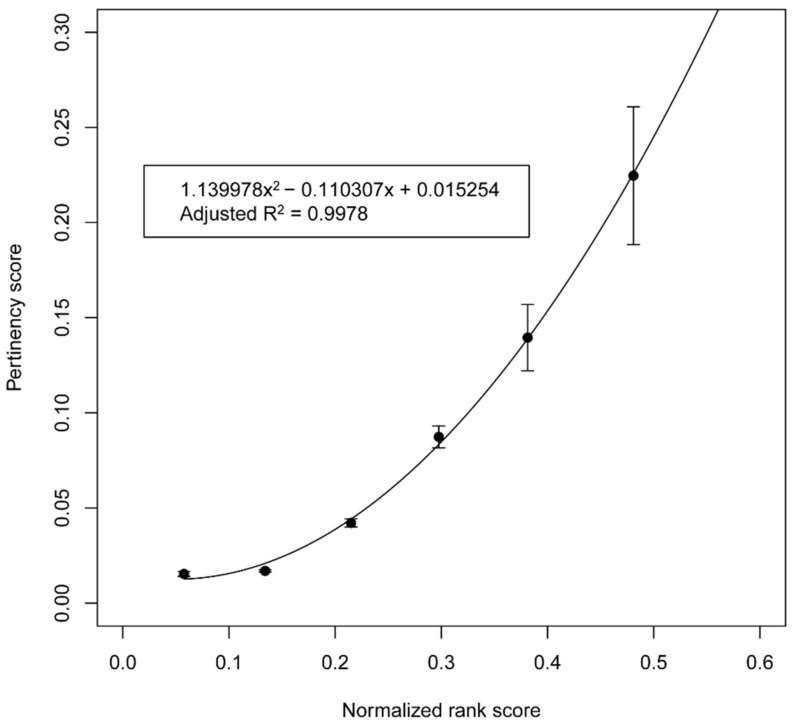
Plot of the pertinency scores versus priority scores. The priority scores of proteins in the protein–disease datasets were collected into six equally spaced intervals along the priority score range. The points on the ordinate of the plot are the means of the fractions of proteins that interact with medications currently known to treat the corresponding disease, termed the pertinency score, for the six folds of the validation set for each of the priority score intervals. The error bars are the standard errors of the means across the six folds. The curved black line indicates the fit using a weighted least squares regression with a second-order polynomial. The resulting equation is y = 1.399x^2^ − 0.110x + 0.015, and the correlation coefficient is 0.9978.

**Figure 3 biomolecules-12-01559-f003:**
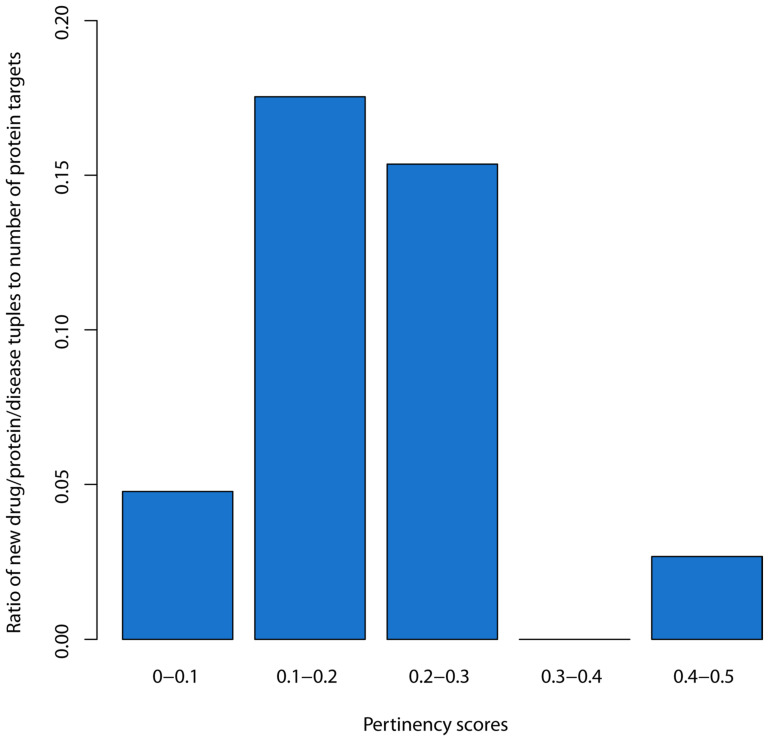
Bar chart of the ratio of the number of drug/protein/disease tuples to the number of proteins targets for each pertinency score bin.

**Table 1 biomolecules-12-01559-t001:** Area under the curve (AUC) values for the retrieval of known targets of the queried diseases. The positives are proteins known to be targeted by medications currently used to treat the disease. The negative comparison set is represented by proteins of the protein–disease dataset that are not known to be targeted by a medication used to treat the disease. The average AUC value per analysis was determined by obtaining the mean AUC when using the 513 diseases of the validation set.

Type of Function(S) Used in The Priority Score Formulation	Average AUC and Standard Deviation	*p*-Value of *t*-Test Relative to 0.5
All	0.68936 ± 0.25888	1.08310 × 10^−49^
All but ChEBI and SUPERFAMILY	0.68661 ± 0.26522	9.66598 × 10^−47^
UniProt keywords only	0.65680 ± 0.25919	1.22968 × 10^−36^
GO molecular function only	0.65522 ± 0.24997	3.12006 × 10^−38^
GO biological process only	0.65329 ± 0.26250	1.42324 × 10^−34^
UniProt residue features	0.62528 ± 0.267669	7.01153 × 10^−24^
GO cellular component	0.61153 ± 0.27507	1.02508 × 10^−18^
InterPro	0.58814 ± 0.00050	3.54596 × 10^−14^
Enzyme commission (EC) number	0.53205 ± 0.25601	0.00570
SUPERFAMILY identifier	0.47983 ± 0.25644	0.07545
ChEBI	0.42085 ± 0.25987	1.55528 × 10^−11^

**Table 2 biomolecules-12-01559-t002:** Illustrative candidates for drug repositioning across all diseases. For each medication, the protein target, the current use, and the putative indication are listed. The estimated pertinency score for the possible indication is also shown.

Medication	Target	UniProt ID	Current Use(s)	Possible Indication(s)	Pertinency Score
Sotorasib	GTPase KRas	P01116	Non-small cell lung cancer with KRAS G12C mutation	Linear nevus sebaceous syndrome	0.5861
Pyrimethamine	Beta-hexosamididase subunit beta	P07686	Toxoplasmosis	GM2 gangliosides	0.4671
Tolcapone	Genome polyprotein	P29990	Parkinson’s disease	Dengue hemorraghagic fever	0.1942

**Table 3 biomolecules-12-01559-t003:** Protein targets or biochemical pathways together with drug repositioning candidates for the possible treatment of Alzheimer’s disease.

Medication	Target/Pathway	Uniport ID	Current Use(s)	Pertinency Score
Insulin	Isoform short of insulin receptor	P06213-2	Types 1 and 2 diabetes mellitus	0.1666
Riluzole	Alpha-synuclein	P37840	Amyotrophic lateral sclerosis	0.1637
Diminazene aceturate (DIZE) to increase ACE2 activity	Neuroinflammation pathway of angiotensin-converting enzyme (ACE1)	P12821	Trypanosomiasis	0.1442

## Data Availability

The results for the study with the timestamp at time of publication are available at https://protein.som.geisinger.edu/Pharmacorank/Downloads/publication_data/. Updates to the results upon further application of the described algorithms are available at the URL https://protein.som.geisinger.edu/Pharmacorank.

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
