# Peer review of "The Pharmacorank Search Tool for the Retrieval of Prioritized Protein Drug Targets and Drug Repositioning Candidates According to Selected Diseases"

_biomolecules, 2022, doi:10.3390/biom12111559_

Round 1
Reviewer 1 Report
The Gnilopyat et al. paper presents a novel search tool named Pharmacorank. The tool, available through a web interface, aims to aid the drug repositioning process. Pharmacorank is based on a novel prioritization algorithm for protein drug targets with satisfying statistical performance. The authors do an excellent job in contextualizing the tool into the drug repositioning ecosystem but, in my opinion, fell a little short when discussing the advantages of their tool with respect to the others.
Because of that, I recommend to ACCEPT the paper after MAJOR REVISIONs.
Major remarks:
* The authors should give a threshold for the end-user to evaluate the pertinency of a gene
* The authors should explicitly affirm the advantages of their tool with respect to other tools in the same ecosystem
Minor remarks:
38-39 I struggle to interpret the author's meaning with "the degree to which they contribute the diseases' etiologies." Can they explain and possibly rephrase more straightforwardly?
45-55 The authors briefly describe here(45-51) two methods: "presence of co-occurring database terms" and "co-occurrences of a diverse set of functional annotation." They then present tools that use "the integration of such methods" (51), giving the idea that they apply different algorithms to multiple databases. Nevertheless, they conclude the paragraph by saying, "there two tools use heterogeneous [...] annotations", suggesting that these methods focus on the INPUT data rather than the algorithm. Could the authors clarify that and possibly rephrase the paragraph to be less ambiguous?
METHODS:
I would advise the authors to maintain a consistent ordering between the summary in the Overview subsection and the presentation of the sections.
100-109 The authors say at the beginning (99-100), "To prepare [...] the titles, abstracts, and [...] (MeSH) terms were obtained for journal articles", but later in the text, they say, "The text fields of the abstracts for these PubMed entries were then retrieved." It is unclear whether they are working on all fields or just the abstract. Also, on line 100, they say, "each SwissProt entry." Are those ALL SwissPros entries, or is that a subset?
110-117 How much do these expansions increase the id population size?
This information could help evaluate the success of the data-mining task.
119 This section is presented as step three in the overview summary section. Why is here presented as the second one?
172 "specific function of the given type of function" is redundant. Instead, consider using just "specific function."
176: use "dot product" instead of "product" and "1d array" or "scalar" instead of "array."
183-184 This is redundant since the authors previously gave (175-177) that exact textual definition of pscore.
188-190 I do not understand why the author chose to add this technical bit.
231 Notation for beta is ambiguous: is the parameter function_family-dependent, or not?
Figure 1: Please update the figure description based on my consideration of the Method section. Finally, it seems there are also issues with the "UN_keywr." FuncType: it should be written as superscript rather than subscript (based on the main text notation).
245-247 How do the authors generate these ROC curves? Are they using R? Please give some context (as they do with the poly fit)
267-268 May I ask the authors to share the function they used to perform the fit?
Results
295: The authors wrote "three," but I think they mean "two."
Table 1: By a naive calculation, I assume that the AUC for ~ 30% of diseases is below .5. It would be interesting to see if these diseases share something.
Reviewer 2 Report
In their manuscript Gnilopyat et al. presented an online search tool for proteins associated with a particular disease along with a list of known drugs as well as compounds which are potentially worth exploring for repurposing. They combined the list of diseases with the linked proteins from multiple sources, such as journal abstracts and other databases like DO, KEGG, OMIM, and DisGeNET for diseases and MEDI, DrugCentral, and ChEMBL for drugs. Moreover, they created and validated a pertinency score to rank the proteins according to their contribution to a particular disease. The manuscript is well written, however some small issues could be seen in the Results and Discussion sections.
However, I got some comments and suggestions regarding the manuscript.
- In the Section 3.3 the authors presented a list of compounds which could be of high success of repurposing. However, the mechanism of the repurposing is way oversimplified in that section. While the idea of repurposing is that some drugs could be used against more than one disease, it is not so straightforward. The potential drug still requires additional clinical tests to obtain additional information on the dosage and an overall mechanism. Moreover, some diseases are more like a spectrum of symptoms rather than an actual "0"-"1" situation, in which 0 stands for the lack of the disease and 1 for the disease. Otherwise, there will be no patients with diabetes who suffer from Parkinson's disease. I think the authors should stress that in the manuscript.
- In the case of cystic fibrosis, the highest ranked protein is the CFTR protein from mouse. That is probably due to the amount of results published which were done using this protein and not the actual human CFTR protein. I suggest adding the source of the organism or perhaps merge the same results obtained for the same protein but with other origin.
- The authors could also consider adding a short description of the protein (for example using the Function section from UniProt) and the disease (Definition section in DO). In my opinion it would be helpful for the user to assess (besides using the pertinency score) if a particular protein is related to the disease.
- The manuscript is not balanced. The authors explained well the implementation of the webserver, its features and validation, but they did not put so much effort in analyzing the results. Given the title of the journal the manuscript was sent, I recommend the authors to rewrite the Results and Discussion sections in order to present the potential applications of the webserver as well as putting more stress on the "bio" perspective.
- I noticed some technical problems with the webserver - in some cases I couldn't connect with the site, even using different web browsers or even different web providers.
Round 2
Reviewer 1 Report
The authors revised the paper and all the issues raised have been addressed.
Reviewer 2 Report
The authors have addressed all my comments and concerns.
Good job!